

# Spatiotemporal variability of NO₂ and PM₂.₅ over Eastern China: observational and model analyses with a novel statistical method

Mengyao Liu[1], Jintai Lin[1], Yuchen Wang[1,2], Yang Sun[3], Bo Zheng[4], Jingyuan Shao[1], Lulu Chen[1], Yixuan Zheng[5], Jinxuan Chen[1,6], May Fu[1], Yingying Yan[1], Qiang Zhang[4], Zhaohua Wu[7,8]

[1] Laboratory for Climate and Ocean-Atmosphere Studies, Department of Atmospheric and Oceanic Sciences, School of Physics, Peking University, Beijing 100871, China

[2] Earthquake Research Institute, The University of Tokyo, Tokyo 113-0032, Japan

[3] Institute of Atmospheric Physics, Chinese Academy of Sciences

[4] Center for Earth System Science, Tsinghua University, Beijing 100084, China

[5] Ministry of Education Key Laboratory for Earth System Modeling, Department of Earth System Science, Tsinghua University, Beijing 100084, China

[6] Max Planck Institute for Biogeochemistry, Hans-Knöll-Str.10, 07745 Jena, Germany

[7] Center for Ocean–Atmospheric Prediction Studies, Florida State University, Tallahassee, Florida 32306-2741, USA

[8] Department of Earth, Ocean and Atmospheric Science, Florida State University, Tallahassee, Florida 32306-4520, USA

Correspondence: J.-T. Lin (linjt@pku.edu.cn)

## Abstract

Eastern China (27°N–41°N, 110°E–123°E) is heavily polluted by nitrogen dioxide (NO₂), particulate matter with aerodynamic diameter below 2.5 μm (PM₂.₅) and other air pollutants. These pollutants vary in a variety of temporal and spatial scales, with many temporal scales non-periodic and non-stationary, challenging proper quantitative characterization and visualization. This study uses a newly compiled EOF-EEMD analysis-visualization package to evaluate the spatiotemporal variability of ground-level NO₂, PM₂.₅, and their associations with meteorological processes over Eastern China in Fall-Winter 2013. Applying the package to observed hourly pollutant data reveals a primary spatial pattern representing Eastern China-wide synchronous variation in time, which is dominated by diurnal variability with a much weaker day-to-day signal. A secondary spatial mode, representing north-south opposing changes in time with no constant period, is characterized by wind-related dilution or buildup of pollutants from one day to another.

We further evaluate simulations of GEOS-Chem and WRF/CMAQ in capturing the spatiotemporal variability of pollutants. GEOS-Chem underestimates NO₂ by about $17 \mu g/m^3$ and PM₂.₅ by 35 $\mu g/m^3$ on average. It reproduces the diurnal variability for both pollutants. For the day-to-day variation, GEOS-Chem reproduces the observed north-south contrasting mode for both pollutants but not the Eastern China-synchronous mode (especially for NO₂).



The model errors are due to a first model layer too thick (about 130 m) to capture the near-surface vertical gradient, deficiencies in the nighttime nitrogen chemistry in the first layer, and missing secondary organic aerosols and anthropogenic dusts. CMAQ overestimates the diurnal cycle of pollutants due to too weak boundary layer mixing – especially in the nighttime, CMAQ overestimates $NO_2$ by about 30 μg/$m^3$ and $PM_{2.5}$ by 60 μg/$m^3$. For the day-to-day variability, CMAQ reproduces the observed Eastern-China synchronous mode but not the north-south opposing mode of $NO_2$. Both models capture the day-to-day variability of $PM_{2.5}$ better than that of $NO_2$. These results shed light on model improvement. The EOF-EEMD package is freely accessible.

## 1. Introduction

Eastern China (EC, 25°N–41°N, 110°E–123°E) is heavily polluted by anthropogenic emissions in recent years (Cui et al., 2016; Klimont et al., 2017; Lin et al., 2015; Richter et al., 2005; Zhang et al., 2016b). Pollutants from this region have also raised concerns on long-range transport to downstream areas (Cooper et al., 2010; Jiang et al., 2015; Lin et al., 2014, 2008; Zhang et al., 2014). Since 2013, the Ministry of Environmental Protection (MEP) of China has greatly expanded its air pollution monitoring network to measure hourly near-surface mass concentrations of particulate matter with aerodynamic diameter less than 2.5 μm ($PM_{2.5}$), $PM_{10}$, nitrogen dioxide ($NO_2$), carbon monoxide, ozone, and sulfur dioxide. These measurements have been used for air pollution analyses and model evaluation (Wang et al., 2014; Xie et al., 2015; Zhang et al., 2016b; Zhao et al., 2016).

Over Eastern China, $NO_2$ and $PM_{2.5}$ concentrations vary diurnally and from one day to another. $NO_2$ is short lived (hours), and its diurnal cycle is affected by rush hour traffic emissions (Chen et al., 2015; Hu et al., 2014), other emission sources, planetary boundary layer (PBL) mixing (Lin and McElroy, 2010), and chemistry (Lin et al., 2012). Although previous studies in the US, Germany and Japan have suggested a weekly cycle of $NO_2$ due to variations in industrial/traffic emissions, such an emission-driven weekly cycle is not visible over the developing countries such as China and India (Beirle et al., 2003; Boersma et al., 2009; Cui et al., 2016; Hu et al., 2014; Kaynak et al., 2009). Instead, ground-based observations show that the day-to-day variation in $NO_2$ over China is associated with changes in meteorological parameters such as wind speed, relative humidity (RH), surface pressure and temperature (He et al., 2017; Zhang et al., 2015).

For $PM_{2.5}$ over China, both the diurnal and the day-to-day variations are complicated by its relatively long lifetime, its various components from different sources, and meteorology. Liu (2016) suggested three types of $PM_{2.5}$ diurnal cycle within a year, with the peak concentration occurring at distinctive hours in different seasons. In the summertime (April to August), the diurnal cycle may follow human activities (Gong et al., 2007; Liu et al., 2016), which is different from the diurnal cycles in the biomass burning season or in winter. Other studies suggested weak diurnal cycles of $PM_{2.5}$ in urban or suburban areas (Chen et al., 2015; Hu et al., 2014). Moreover, some studies pointed to lack of weekly cycle of $PM_{2.5}$ (Liu et al., 2016), while others suggested contrasting weekly cycles (for Beijing, Chen et al., 2015; Hu et al., 2014). In winter, the frequent and irregular weather systems prohibit a clear weekly cycle (Gong et al, 2007).





This study analyzes the spatiotemporal variability of $NO_2$ (with the shortest lifetime of hours
and the greatest variability among the pollutants measured by the official monitoring
network) and $PM_{2.5}$ (the dominant pollutant) over Eastern China in Fall-Winter 2013. Given
the complex and non-stationary nature of pollutant variability over Eastern China, here we
compile an EOF-EEMD analysis-visualization package to simultaneously distinguish and
visualize the spatial and temporal variability of pollutants. In sequence, the package consists
of an Empirical Orthogonal Function (EOF) analysis (Lorenz, 1956) to separate spatial and
temporal patterns, an Ensemble Empirical Mode Decomposition (EEMD) analysis (WU et
al., 2009) to separate different temporal modes, a Hilbert Transform (HT), a Marginal
Spectrum Analysis (MSA), and a visualization step to present all physically meaningful
spatial and temporal modes in a two-dimensional plot. In particular, EEMD (Huang, 2005;
Huang et al., 1998, 1999; Huang and Attoh-Okine, 2005; WU et al., 2009) is an effective tool
to extracting signals from noisy nonlinear and non-stationary processes (WU et al., 2009).
EEMD and its variants (e.g., MEEMD) have been widely used in climate studies (Feng et al.,
2014; Huang et al., 2012a, 2012b; Vecchio and Carbone, 2010; Wu et al., 2011, 2016). The
EOF-EEMD package thus allows for quantitative manifestation of the spatial, (regular)
diurnal and (irregular) day-to-day variations of pollutants and meteorological drivers.

We further use the EOF-EEMD package to evaluate how well chemical transport models
(CTMs) can reproduce the observed pollution variability. Although popularly used in air
pollution diagnosis, forecast/projection, and remote sensing (Geng et al., 2015; Lin et al.,
2015), models are subject to errors in emissions, chemistry, transport, PBL mixing and other
processes (Lin et al., 2008, 2012; Zhang et al., 2016b). This study evaluates two
representative models, including GEOS-Chem and WRF/CMAQ, with a note that such
evaluation can be applied to other models.

The rest of the paper is organized as follows. Section 2 introduces in-situ measurements of
$NO_2$, $PM_{2.5}$ and meteorological parameters, model simulations, and the EOF-EEMD analysis-
visualization package. Section 3 analyzes the observed spatiotemporal variations of $NO_2$ and
$PM_{2.5}$, including their relationships with meteorological parameters. Section 4 evaluates the
modeled spatiotemporal variations of $NO_2$ and $PM_{2.5}$. Section 5 concludes the present study
with further discussion on the applicability of the EOF-EEMD package.

## 2. Data and Methods

### 2.1 Spatial and temporal domain

We focus on pollution over Eastern China (25°N–41°N, 110°E–123°E). Guided by an EOF
analysis, we contrast pollution over the southern (SEC, south of 35°N) and northern (NEC)
parts to address the regional differences in day-to-day pollution variability. Such latitudinal
separation coincides with the Huaihe River climate transitional zone (Ye and Li, 2017). The
orange line in Fig. 1 separate the two regions.

Our study period is from October 25$^{th}$ to December 25$^{th}$ 2013, with a total of 1488 hours in
62 days. Most air pollution data are missing in January and February 2014 because of



instrumental failure or data retrieval failure; and data before October 25$^{th}$ are not available.

*2.2 $NO_2$ and $PM_{2.5}$ observations*

We retrieve hourly measurements of $NO_2$ and $PM_{2.5}$ from 193 air quality monitoring stations of the MEP. Most stations are located in the urban areas, and only six stations are suburban. As almost every station has missing values in more than one day, we exclude stations that have missing values at $\geqslant$ 30% of the 1488 hours or during consecutive 72 hours. We thus select 163 stations for $NO_2$ and 159 stations for $PM_{2.5}$ in the same 42 cities. The dots in Fig.
1a, b depict the stations and cities, respectively. The blue dots show stations/cities with both valid $NO_2$ and $PM_{2.5}$, the green dots with $NO_2$ only, and the purple dots with $PM_{2.5}$ only. The slight difference between $NO_2$ and $PM_{2.5}$ stations does not affect our analysis of the regional pattern of pollutants.

*2.2.1 Correction of raw $NO_2$ measurements*

At the monitoring sites, $NO_2$ is measured via molybdenum-catalyzed conversion to nitric oxide (NO) and a subsequent chemiluminescence measurement. The measurement technique suffers from interference by more oxidized nitrogen species, since the heated molybdenum surface exhibits low chemical selectivity (Boersma et al., 2009; Lamsal et al., 2008; Zhang et al., 2016a)

Here we follow Lamsal et al. (2008) to correct for the interference, by introducing a correction factor (CF) based on GEOS-Chem simulated nitrogen species ($NO_2$, $HNO_3$, PAN and all alkyl nitrates ($\sum AN$)):

$$CF = \frac{NO_2}{NO_2 + \sum AN + 0.95PAN + 0.35HNO_3} \qquad (1)$$

We multiply CF with the raw $NO_2$ data to obtain "corrected" $NO_2$ concentrations. Our
sensitivity test suggests that assuming PAN and $HNO_3$ to be fully converted to $NO_2$ (i.e., assuming the coefficients to be unity for both PAN and $HNO_3$ in Eq. 1) does not affect our spatiotemporal analysis of $NO_2$. Hereafter the $NO_2$ "corrected" by Eq. 1 is discussed, unless stated otherwise.

Figure 2 compares the regional mean hourly time series of raw and "corrected" $NO_2$. The
correction reduces $NO_2$ concentrations by about 2–30 μg/$m^3$ over the whole period, and is higher at times when nitrogen is more oxidized. It slightly reduces the relative contribution of day-to-day variability to the total variance of $NO_2$ under the EOF-EEMD analysis (not shown), because excluding those more oxidized species shortens the lifetime of $NO_2$.

*2.2.2 Filling in missing values for EOF-EEMD analysis*

Prior to an EOF-EEMD analysis, we fill in missing values in hourly pollution observations. If data are missing for more than 12 consecutive hours, we fill in the missing value in each hour with data on that hour averaged over all days; as such, the diurnal cycle is maintained. In



other cases, linear interpolation from adjacent valid data is applied. Our interpolation does not introduce significant artificial information for spatiotemporal analysis, as validated by a
sensitivity test with GEOS-Chem model data. Specifically, the EOF-EEMD results based on the original GEOS-Chem data (i.e., no missing values) are similar to the results based on model data sampled at times of valid observations with missing values filled in the same manner as for the observation data.

### 2.2.3 Conversion from station- to city-based datasets

Since different cities have different numbers of stations, we calculate city mean observations by averaging across all stations of each city. Compared to a station-based analysis, the city-based EOF-EEMD results reduce the spatial noises leading to more distinctive temporal patterns. All analyses hereafter is based on city mean data. The longitude/latitude of each city center is used to identify respective model grid cell.

### 2.3 Meteorological observations

We use 3-hourly measurements of 2-meter air temperature, 2-meter relative humidity and 10-meter wind speed from meteorological stations recorded at the National Oceanic and Atmospheric Administration National Centers for Environment Information (NOAA NCEI). The locations of these stations do not always coincide with air pollution stations. Thus, we
select 36 meteorological stations within 10 km of air pollution stations (red hollow dots in Fig. 1). Despite the difference (in number and location) between pollution and meteorological stations, an analysis of the regional-temporal patterns of pollutants and meteorology is still informative (see Sect. 3.2).

To fill in missing values, we apply an interpolation process that accounts for diurnal
variability, using information in an adjacent day. For example, if temperature on October 26$^{th}$ 12:00 is missing, we calculate the temperature difference between 9:00 and 12:00 on 25$^{th}$ as well as the difference between 15:00 and 12:00 on 25$^{th}$. We then use these differences to adjust the temperatures at 9:00 and 15:00 on 26$^{th}$, respectively, and finally use the mean of the two adjusted temperatures as the temperature on 26$^{th}$ 12:00.

For consistency with the hourly pollution data, we linearly interpolate the 3-hourly meteorological measurements to each hour. This interpolation does not distort the EOF-EEMD analysis, as confirmed by comparing the statistical analysis on 1-hourly GEOS-FP meteorological parameters versus an analysis on 3-hourly GEOS-FP data. Note that the GEOS-FP meteorology is used to drive GEOS-Chem.

### 2.4 Model simulations

### 2.4.1 GEOS-Chem

We use the nested GEOS-Chem CTM version 9-02 (Zhang et al., 2016a) to simulate $NO_2$, $PM_{2.5}$ and other pollutants over China in October–December 2013. The model resolution is 0.3125° long. × 0.25° lat. grid with 47 vertical layers, and the lowest 10 layers are of ~ 130 m





thickness each. The model is driven by the GEOS-FP assimilated meteorology from the National Aeronautics and Space Administration (NASA) Global Modeling and Assimilation Office, with the full $O_x$-$NO_x$-VOC-CO-$HO_x$ gaseous chemistry (Mao et al., 2013) and online aerosol calculations. Vertical mixing in the PBL adopts a non-local scheme (Holtslag and Boville, 1993; Lin and McElroy, 2010). Model convection is simulated with the Relaxed Arakawa-Schubert scheme (Rienecker et al., 2008).

Chinese anthropogenic emissions of $NO_x$ and other pollutants adopt the monthly MEIC inventory with a base year of 2010 (www.meicmodel.org) (Geng et al., 2017). We further use the monthly DOMINO v2 $NO_2$ data to scale monthly anthropogenic $NO_x$ emissions from 2010 to the simulation year (Lin et al., 2015). The emission scaling improves the simulation of $NO_2$ (Cui et al., 2016). Other model setups are referred to Lin et al. (2015) and Yan et al. (2016).

GEOS-Chem modeled $PM_{2.5}$ includes secondary inorganic aerosols (sulfate, nitrate and ammonium), black carbon, primary organic carbon, natural dusts, and sea salts. Secondary organic aerosols are not included, considering the severe underestimate in China due to missing precursor emissions and formation pathways (Fu et al., 2012; Zhang et al., 2016a). Anthropogenic dusts are also not included.

The nested model simulation is from 15$^{th}$ October to 25$^{th}$ December in 2013, allowing for a 10-day spin-up period. Its lateral boundary conditions of chemicals are updated every 3 h by results from a corresponding global simulation on a 2.5° long. × 2° lat. grid. Modeled $NO_2$ and $PM_{2.5}$ in the first layer are sampled at city centers and times with valid observations, unless stated otherwise.

*2.4.2 CMAQ*

We use the Weather Research and Forecasting (WRF) model v3.5.1 (http://www.wrf-model.org/) to drive CMAQ v5.0.1 (http://www.cmascenter.org/cmaq/). The simulation covers East Asia at a horizontal resolution of 36 × 36 km$^2$ with 14 vertical layers. The lowest six layers are of ~ 80 m thickness each, and about eight layers are below 1 km. The gas-phase chemistry uses the CB05 mechanism with active chlorine chemistry and updated toluene mechanism (Whitten et al., 2010). The aqueous-phase chemistry adopts the updated Regional Acid Deposition Model (RADM) (Chang et al., 1987; Walcek and Taylor, 1986). The aerosol chemistry follows AERO6. PBL mixing in both WRF and CMAQ adopts the ACM2 scheme (Pleim, 2007). Other model physics are detailed in Zheng et al. (2015).

Chinese anthropogenic emissions are from MEIC (www.meicmodel.org). Emissions in 2013 are extrapolated from the base year (2012) based on country-level statistics (Zheng et al., 2015). Anthropogenic emissions in other Asian countries and biomass burning emissions are taken from the MIX emission inventory prepared for the Model Inter-Comparison Study Asia Phase III (MICS-ASIA III).



The PM$_{2.5}$ species in AERO6 include fine mode sulfate, nitrate, ammonium, primary and secondary organic aerosols, black carbon, sodium, calcium, aluminum, particulate chloride, and remaining unspeciated fine mode primary PM

(http://www.airqualitymodeling.org/cmaqwiki/index.php?title=CMAQv5.0_PMother_speciation).

The simulation is from 15$^{th}$ October to 25$^{th}$ December 2013, allowing for a 10-day spin-up period. Initial conditions and boundary conditions are from GEOS-Chem (Zheng et al., 2015). Modeled NO$_2$ and PM$_{2.5}$ in the first layer are sampled at city centers and times with

valid observations, unless stated otherwise.

*2.5 EOF-EEMD analysis-visualization package*

As shown in Fig. 3, our EOF-EEMD analysis-visualization package consists, in order, of an EOF analysis (Lorenz, 1956), an EEMD analysis (WU et al., 2009), a Hilbert Transform (HT) with Marginal Spectrum Analysis (MSA), and a visualization step to quantitatively depict the

spatial-temporal scales of measurement or model data.

The basic purpose of our package is to quickly and simultaneously identify and visualize various spatial and temporal scales of interest in the observation or model datasets. As shown by Feng et al. (2014) and Wu et al. (2016), combining EOF with EEMD to decompose the datasets leads to a faster calculation than MEEMD by one or two orders, because here the

EEMD is applied to the temporal components (i.e., PCs) out of an EOF analysis rather than to all dimensions. Also, our EOF-EEMD package conducts additional HT-MSA and provides visualization of all spatial and temporal scales of interest.

- EOF analysis to decompose a two-dimensional dataset (time series at multiple locations) into spatial and temporal components.

Suppose there are $n$ locations, each having a time series of length $p$. The associated dataset **Z** is an n×p matrix. An EOF analysis of **Z** gives:

$$\mathbf{Z} = \mathbf{U} \sum \mathbf{W^T} \qquad (2)$$

Here **Σ** is a diagonal q×q matrix containing the first q singular values of **Z**, and it represents the contribution of each pattern to the total variance of **Z**. The diagonal values of **Σ** is in a

descending order, thus the first several modes are the dominant ones. **U** is an n×q matrix representing the spatial component, and each column of **U** represents a spatial mode. **W** is a p×q matrix representing the temporal component, and each column of **W** represents a principle component (PC) for temporal variation associated with the corresponding spatial mode.

- EEMD analysis of each PC time series to obtain its "intrinsic mode functions" (IMFs) of descending frequencies.





Each PC is mixed with multiple scales, which requires further decomposition in the time domain. Unlike Fast Fourier Transform (FFT) or Wavelet Transform (WT), EEMD does not need priori bases, and it can be appropriately applied to delineate nonlinear and non-stationary time series, as in our pollution study.

EEMD consists of an ensemble of Empirical Mode Decomposition (EMD) performed on each PC time series (denoted as $x(t)$ in Eq. 3). Each EMD linearly decomposes $x(t)$ into individual IMFs $c_j$ (of ascending time scales and descending frequencies) and a residual $r_n$:

$$x(t) = \sum_{j=1}^{n} c_j(t) + r_n(t) \qquad (3)$$

EMD is based on finding local maxima and minima of the time series. A detailed decomposition process can be found in Huang et al. (1998, 1999). EMD is much less susceptible to missing values and data interpolation than approaches that are based on an analysis of the whole time series (e.g., FFT and WT).

EMD may be sensitive to noise in the real data to encounter a "mode mixing" problem (WU et al., 2009). EEMD solves this problem by performing an ensemble of hundreds of EMDs, each with certain white noise added to $x(t)$. Hence, the noise in the real data is incorporated as part of the while noise, and the ensemble further minimize the effects of noise. The white noise is assumed to follow the standard Gaussian distribution (WU et al., 2009). Figure 4 shows an example of the EEMD analysis.

- Hilbert Transform and Marginal Spectrum Analysis of each IMF to reveal its representative frequency range.

There are no discrete periods/frequencies in the pollution and meteorological time series. Correspondingly, an IMF also has a continuous frequency range (rather than a constant frequency) that can be determined by HT-MSA. The HT reveals the IMF's energy-frequency-time distribution (Huang et al., 1999). The MSA further shows the IMF's distribution of variance (energy) with respect to different frequencies. The spectral peak represents the largest contribution to total variance.

A spurious oscillation may occur near the edges of certain IMF time series, resulting in an inaccurate calculation of variance under HT-MSA. We apply a box-car filter (Gubbins, 2004) to select the internal 60% of an IMF time series (from 20% to 80% of the 1488 hours) to perform HT-MSA. Figure 4b shows an example of the visualized result of HT-MSA, where the horizontal axis is the number of occurrences within the whole period (frequency, in $h^{-1}$, multiplied by the time length, 1488 hours) and the vertical axis the energy contribution. IMF2 ~ IMF5 are visualized and analyzed in this study. The higher-frequency IMF1 is noisy as the energy is distributed over a wide range of occurrence numbers. IMF6 ~ IMF10 represent the longest temporal scales that contribute little to the total variance of the decomposed PC. Thus IMF1 and IMF5~IMF10 are not further analyzed.

Based on HT-MSA, we determine a representative frequency range (RFR) such that the range encompasses the peak frequency, and that the frequencies within the range contribute 50% of



the total variance of an IMF. The frequencies below and above the RFR bounds each contribute to 25% of the total variance of the IMF. Before calculating the RFR, we smooth the marginal spectrum by connecting all local maxima of the spectrum with cubic spline.

-    Visualization of the spatial and temporal scales in a two-dimensional plot.

Finally, we simultaneously visualize the spatial and temporal scales as well as their
contributions to the total variance of **Z** in a two-dimensional plot, for easy observational diagnosis and model evaluation. In this plot, an IMF is represented by a vertical "error bar" and a horizontal bar. The length of the "error bar" stands for the representative period range (RPR, the inverse of RFR), and a shorter length means a more stationary variation mode (i.e., towards a fixed frequency/period). The length of the horizontal bar stands for the contribution
to the total variance. For clearer presentation, the plot does not include IMFs which do not pass the white noise examination, which lay outside the range of scales considered here (hours to days), or which contribute little to the total variance of the original data (e.g. less than 1%).

### 3. Observational analyses of $NO_2$, $PM_{2.5}$ and meteorological variables

*3.1 General characteristics*

The colored dots in Fig. 5a, b show the observed spatial distributions of city-mean $NO_2$ and $PM_{2.5}$ averaged over the time period. Both $NO_2$ and $PM_{2.5}$ are largest over Beijing-Tianjin-Hebei (BTH) in the north and the Yangtze River Delta (YRD) in the east. $NO_2$ concentrations exceed 60 µg/$m^3$ at many sites. The range of $PM_{2.5}$ is larger, from below 10 µg/$m^3$ in some
northern and coastal cities to about 200 µg/$m^3$ in several cities of BTH.

Figure 6a, b shows the diurnal variations of $NO_2$ and $PM_{2.5}$ over Eastern China, NEC and SEC, averaged over all days. Similarly, over the three regions, $NO_2$ peaks around 19:00 due to evening rush hour emissions, reduced PBL mixing, and a lengthened lifetime. $NO_2$ reaches a minimum at 14:00 because of the shortest lifetime and strongest PBL mixing. The diurnal
range (maximum minus minimum) is about 30 µg/$m^3$. The $PM_{2.5}$ level also reaches a minimum in the early afternoon. It has a much smaller diurnal range at 10 µg/$m^3$. The vertical error bars in Fig. 6a, b depict the standard deviation for day-to-day variation of $NO_2$ and $PM_{2.5}$ at any given hour. At a given hour, the $PM_{2.5}$ level is much more variable across the days than $NO_2$. In particular, the day-to-day standard deviation for $PM_{2.5}$ at a given hour is as
large as the diurnal range of $PM_{2.5}$.

Figure 6c, d further shows the time series of daily mean $NO_2$ and $PM_{2.5}$. All data are de-trended (trends are at 0.01 µg/$m^3$/$hour$ for $NO_2$ and 0.05 µg/$m^3$/$hour$ for $PM_{2.5}$). Although local maxima and minima (peaks and troughs of the time series) occur every several days, there is no single period or amplitude for the variation of each species. For $NO_2$
over Eastern China (black line in Fig. 6c), the local maxima vary from 60 to 100 µg/$m^3$, and the local minima vary from 20 to 40 µg/$m^3$. For $PM_{2.5}$ over Eastern China (black line in Fig. 6d), the local maxima vary from 100 to 300 µg/$m^3$, and the local minima vary from 20 to 120 µg/$m^3$. Furthermore, comparing the green and blue lines reveals that pollutants over NEC and SEC synchronize in some days but are out of phase in others; this feature is
quantitatively analyzed in Sect. 3.2. These day-to-day variation patterns are associated with meteorological conditions and pollutant lifetimes.



Figure 7 shows day-to-day anomalies of observed pollutant concentrations and meteorological parameters over NEC and SEC. All data are de-trended. Over NEC, wind speed is clearly anti-correlated with pollutant levels. The correlation coefficient reaches -0.73 between $NO_2$ and wind speed and -0.60 between $PM_{2.5}$ and wind speed. Over this region, stronger winds are often associated with lower RH and lower temperature, characteristic of cold air passage that brings cleaner, colder and drier air from the north to NEC and transport the NEC pollution out of the region. Correspondingly, RH is strongly positively correlated with $NO_2$ (R = 0.62) and $PM_{2.5}$ (R = 0.69). The meteorology-associated day-to-day variability is more apparent after Mid November, when the variations of the two pollutants are more synchronous.

Over SEC (Fig. 7), the relationship between pollutant levels and meteorological parameters is more complex. The correlation between daily mean $PM_{2.5}$ and wind speed is relatively weak (R = -0.44, compared to -0.60 over NEC), and its correlation with RH is even weaker (R = 0.29). This indicates that the northerly air does not reduce $PM_{2.5}$ levels over SEC as effectively as over NEC, as $PM_{2.5}$ from NEC may be transported to SEC. By comparison, $NO_2$ is still highly anti-correlated with wind speed (R = -0.77) over SEC, likely a result of the short lifetime of $NO_2$. Compared to $PM_{2.5}$ whose lifetime is sufficiently long (several days) for transport from NEC to SEC (Hu et al., 2014), $NO_2$ has a much shorter lifetime (below one day; Lin et al., 2012) and cannot undergo effective long-distance transport. However, almost all pollution measurement sites are urban, and weaker (stronger) winds allow for rapid accumulation (removal) of urban $NO_2$ pollution.

*3.2 EOF-EEMD analyses of pollutants and meteorological parameters*

Although informative, the time series analyses on regional mean pollution in Sect. 3.1 do not provide adequate quantitative information of the spatiotemporal variability and embedded scales. In fact, the separate discussion on NEC and SEC in Sect. 3.1 is largely inspired by the following EOF-EEMD analysis that suggests distinctive features between these two sub-regions. In this section, we use the EOF-EEMD package to distinguish and visualize the quantitative contributions of individual spatial and temporal modes to variations in the pollutant and meteorological data.

The columns of Fig. 8 show the EOF-EEMD results for the observed temperature, RH, wind speed, $NO_2$ and $PM_{2.5}$, respectively. The first two rows show the first two spatial patterns (EOF1 and EOF2) out of the EOF analysis. The third row visualizes the EEMD-HT-MSA results for PC1 and PC2, the temporal counterparts of EOF1 and EOF2. For all variables, the first two PCs contribute more than 50% of the total variance of the original data. The following PCs (PC3, PC4…) contain small variances and are not discussed here.

*3.2.1 EOF-EEMD analyses of pollutants*

The fourth column of Fig. 8 for $NO_2$ shows a primary pattern (EOF1 and PC1) with synchronous variation over the entire Eastern China. This pattern contributes 42% of the total variance of $NO_2$. The two dominant IMFs of PC1 have time periods at 24 hours and 12 hours, respectively, and they together contribute 30.4% of the total variance of $NO_2$. Thus, PC1 mainly reflects the diurnal variation of $NO_2$. PC1 also contains some day-to-day variability IMFs, which contribute about 10% of the total variance of $NO_2$. The second pattern (EOF2) of $NO_2$ reveals opposite temporal variations between NEC and SEC. This temporal contrast is mainly reflected in the day-to-day variability, with RPRs around 2–5 days contributing 10.9% of the total variance in $NO_2$. The day-to-day components of PC1 and PC2 correspond to the



finding in Sect. 3.1 that $NO_2$ over NEC and SEC are synchronous in some days but out of phase in others.

We further investigate the physical meanings of PC1 and PC2 for $NO_2$. The red solid and red
dashed lines in Fig. 6a, c show the diurnal and day-to-day variations of PC1 and PC2, in comparison to regional mean $NO_2$ levels over Eastern China (black line), NEC (green line) and SEC (blue line). Table 1 shows the associated correlation coefficients. PC1 is synchronous with Eastern China mean $NO_2$ for both diurnal and day-to-day variations (R reaches 1.0), confirming this regionally synchronous pattern. The day-to-day variation of PC2
is correlated to NEC $NO_2$ (R = 0.66) but anti-correlated to SEC $NO_2$ (R = -0.45, Table 1), again confirming this NEC-SEC contrasting pattern.

The last column of Fig. 8 shows the EOF-EEMD result for $PM_{2.5}$. As for $NO_2$, EOF1 and PC1 of $PM_{2.5}$ reflect a temporally synchronous pattern over Eastern China, which contributes 44% of the total variation of $PM_{2.5}$. Again, PC1 is synchronous to Eastern China mean $PM_{2.5}$
(red versus black lines in Fig. 6b, d) in terms of both diurnal and day-to-day variations, with correlation coefficients approaching 1.0 (Table 2). However, the IMFs of PC1 representing diurnal variation are relatively weak, consistent with the noisy diurnal cycle of $PM_{2.5}$ discussed in Sect. 3.1. The dominant IMF of PC1 shows a period around seven days. PC2 of $PM_{2.5}$ reflects the day-to-day contrast between NEC and SEC (Fig. 6d and Table 2) with
RPRs of 2–5 days, similar to PC2 of $NO_2$.

*3.2.2 EOF-EEMD analyses of meteorological parameters*

For comparison, the first three columns of Fig. 8 show the EOF-EEMD results for the observed temperature, RH and wind speed. The EOF-EEMD result for wind speed (the third column in Fig. 8) is closest to that for $NO_2$, with a regionally synchronous pattern (EOF1 and
PC1), a NEC-SEC contrasting pattern (EOF2 and PC2), and a dominant IMF with a period of 24 hours. The day-to-day wind speed variability is also reflected in the IMFs of PC1 and PC2 with RPRs of 2–5 days, consistent with that for $NO_2$. The EOF-EEMD result for wind speed is also fairly comparable with that for $PM_{2.5}$, although the latter shows a dominant IMF (in PC1) with a period of seven days. These results are consistent with Sect. 3.1 but with a more
quantitative analysis on the spatiotemporal scales.

The EOF-EEMD analysis for temperature (the first column of Fig. 8) shows that PC1 contributes 88% of the total variance, and it is dominated by the IMF with a period at 24 hours. The contribution of PC2 is negligible (4%). For RH (the second column of Fig. 8), PC2 plays a minor role, and there are IMFs of PC1 with periods near 3 and 12 days,
contributing to the correlation between RH and $PM_{2.5}$. These results indicate complex association in the day-to-day variability between temperature/RH and pollutants, broadly consistent with the discussion in Sect. 3.1.

## 4. Evaluation of GEOS-Chem and WRF/CMAQ simulations

*4.1 General evaluation*

The color contours in Fig. 5a-d show the horizontal distributions of $NO_2$ and $PM_{2.5}$ simulated by GEOS-Chem and CMAQ. The model results here are averaged from all days over the time





period rather than sampled from days with valid observations. Both models capture the general spatial patterns of observed $NO_2$ and $PM_{2.5}$, with heaviest pollution over the north and east.

Figure 9 evaluates the regional mean diurnal and day-to-day variations of modeled pollutant levels over NEC and SEC. Here model data are sampled from days and locations with valid observations. All trends are negligible and have been removed, consistent with the observational analysis. GEOS-Chem underestimates the observations by about 17 µg/$m^3$ over Eastern China (21 µg/$m^3$ over NEC and 13 µg/$m^3$ over SEC) for $NO_2$ and by 35 µg/$m^3$

over Eastern China (31 µg/$m^3$ over NEC and 41 µg/$m^3$ over SEC) for $PM_{2.5}$ averaged over the whole period. The model bias is relatively consistent across individual hours. GEOS-Chem captures the observed diurnal variability for both pollutants as well as the day-to-day variability of $PM_{2.5}$, although it greatly underestimates the day-to-day variability of $NO_2$. More model evaluation statistics is shown in Table 3.

Figure 9 also shows that WRF/CMAQ overestimates the nighttime observations by about 30 µg/$m^3$ for $NO_2$ and 60 µg/$m^3$ for $PM_{2.5}$ averaged over Eastern China, although it reproduces the daytime pollutant levels. This means an overestimate of the diurnal range, as is also revealed by the EOF-EEMD analysis in Sect. 4.2. CMAQ captures the day-to-day variability of daily mean $NO_2$ and $PM_{2.5}$ much better than GEOS-Chem (R = 0.63–0.84 versus 0.25–

0.37 over NEC and SEC for $NO_2$; and 0.87–0.88 versus 0.55–0.75 for $PM_{2.5}$). Note that the correlations showed here mainly reflect the model capabilities in capturing Eastern China-wide synchronous day-to-day variation; and they do not imply the model performance in simulating the NEC-SEC contrast, which are revealed in Sect. 4.2. More model evaluation statistics is shown in Table 3.

*4.2 Model evaluation based on the EOF-EEMD analysis*

Figures 10 and 11 evaluate the EOF-EEMD results for modeled $NO_2$ and $PM_{2.5}$, respectively. Prior to the EOF-EEMD analysis, modeled $NO_2$ and $PM_{2.5}$ are sampled at times and locations with valid observations and then underwent the same interpolation procedure to fill the missing values. In these figures, the last three rows visualize the EOF-EEMD-HT-MSA

results in different ways (manifested in different lengths of the horizontal bar for each IMF). In the third row, the variance of each IMF is normalized to the total variance of the original data ($NO_2$ or $PM_{2.5}$). In the fourth row, the variance of each IMF is normalized to the variance of its respective PC, in order to better visualize the signals from PC2 (which has a much smaller variance than PC1); as such, only the IMFs from the same PC are intercomparable.

The fifth row visualizes the absolute variance of each IMF without any normalization.

The first two rows of Fig. 10 show EOF1 and EOF2 of $NO_2$. Both GEOS-Chem and CMAQ exhibit a synchronous pattern (EOF1) and a NEC-SEC contrasting pattern (EOF2), consistent with the observation. However, the CMAQ simulated NEC-SEC contrast in EOF2 is much weaker than the observed. Table 1 shows that for modeled $NO_2$, PC1 is highly correlated to Eastern China mean $NO_2$ for diurnal (R = 1.0 for GEOS-Chem and CMAQ) and day-to-day

(R = 0.56–0.97) variability, and that PC2 is correlated to NEC $NO_2$ (R = 0.74–0.81) and anti-



correlated to SEC NO$_2$ (R = -0.47 – -0.32) in terms of day-to-day variability, in line with the observational analysis.

The last three rows of Fig. 10 show that both models underestimate the contribution of day-to-day variability to the total variance of NO$_2$ (with a shorter length of horizontal bar). For PC1, CMAQ captures the RPR (position of "error bar") and variance (length of horizontal bar) of the observed IMFs fairly well. By comparison, GEOS-Chem underestimates the day-to-day variance (too small horizontal length) and does not capture its RPR. These results are consistent with the analysis in Sect. 4.1 (Fig. 9) that CMAQ is correlated with the observed

Eastern China-wide synchronous NO$_2$ time series much better than GEOS-Chem. For PC2, which reflects the NEC-SEC contrasting pattern, GEOS-Chem outperforms CMAQ in capturing the RPR and variance of the observed day-to-day IMFs (red colored in fourth row). This model characteristic is not seen from the time series discussion in Sect. 4.1.

Figure 11 shows that both GEOS-Chem and CMAQ capture the synchronous pattern (EOF1)
and the NEC-SEC contrasting pattern (EOF2) of PM$_{2.5}$. For PC1, GEOS-Chem captures the variance of each IMF but not its RPR (especially for the day-to-day IMFs). CMAQ simulates too strong diurnal IMFs, consistent with its overestimated diurnal cycle discussed in Sect. 4.1. CMAQ outperforms GEOS-Chem in capturing the RPR of day-to-day IMFs of PC1, in line with its better correlation to the observations (Fig. 9). For PC2, GEOS-Chem captures
the variance and RPR of the observed day-to-day IMFs better than CMAQ.

*4.3 Discussion on model deficiencies*

WRF/CMAQ overestimates the diurnal variation of NO$_2$ and PM$_{2.5}$. The causes are multifaceted. The ACM2 PBL mixing scheme in WRF v3.5.1 and CMAQ v5.0.1 (used here) assumes the same value of for eddy diffusivity of momentum (K$_m$) and heat (K$_h$), which
implies a Prandtl number (Pr = K$_m$/K$_h$) of unity and too weak mixing under stable atmospheric conditions (i.e., at night). This deficiency has been alleviated in WRF v3.7 and CMAQ v5.1. Also, there is inconsistency between CMAQ and WRF in the Monin-Obukhov length in the surface layer module. This error has been corrected in CMAQ v5.1. For more model    update    details,    please    refer    to    the    online    document
(https://www.airqualitymodeling.org/index.php/CMAQ_version_5.1_(November_2015_relea se)_Technical_Documentation#Asymmectric_Convective_Model_version_2_.28ACM2.29).

GEOS-Chem (the first model layer) underestimates surface NO$_2$ by about 17 μg/$m^3$ and PM$_{2.5}$ by 35 μg/$m^3$ averaged over Eastern China. The underestimate of PM$_{2.5}$ is in part because GEOS-Chem does not include secondary organic aerosols, which likely contribute as
much as 21% of PM$_{2.5}$ over Eastern China (Fu et al., 2012). Also, the model does not include anthropogenic dusts. Furthermore, although the observation stations are close to the ground, the first layer of GEOS-Chem is too thick (130 m) to fully capture the vertical gradient of pollution concentrations. Figure 12 shows Eastern China mean vertical profiles of NO$_2$ in the two models. The center of the first layer of CMAQ (40 m) is closer to the ground, and the
center of its second layer is located at a height similar to the center of the first layer of GEOS-Chem. CMAQ shows a strong vertical gradient of NO$_2$ from its first to second layer.



Had we used the CMAQ-simulated ratio of the first over second layer to extrapolate GEOS-Chem first-layer $NO_2$ to 40 m, this would significantly increase the model's "ground-level" $NO_2$ (by 24% over NEC and 17% over SEC) and $PM_{2.5}$ (by 45% and 17%). However, the
510 extrapolation does not improve the day-to-day correlation to the observations, indicating the important roles played by other factors. See Table 3 for more evaluation statistics.

GEOS-Chem (the first model layer) also underestimates the Eastern China-wide synchronous day-to-day variation of $NO_2$. When averaged over the 10 lowest layers (below 850 hPa), GEOS-Chem $NO_2$ captures the day-to-day variability of observed surface $NO_2$. This suggests
515 that the model deficiency in day-to-day variability may be specific to the first layer. Moreover, the first layer of GEOS-Chem captures the day-to-day variation of observed $NO_2$ in the afternoon (12:00–15:00 local time, R = 0.9 over NEC and 0.8 over SEC), but the model performance is rather poor in the evening (20:00–23:00 local time, R = 0.1 over NEC and SEC), suggesting nighttime-specific model inadequacies. A further analysis on nighttime
520 ozone and the NO : $NO_2$ ratio suggests that GEOS-Chem greatly underestimates the observed nighttime ozone by 49.2% on average over NEC and 54.6% over SEC, particularly on days when their NO : $NO_2$ ratio are much greater than CMAQ-modeled ratio. The mean NO : $NO_2$ ratio in GEOS-Chem is 1.8 over NEC and 1.4 over SEC, greater than the ratio in CMAQ (1.0 over NEC and 0.4 over SEC) by a factor of 2–3. Overall, it appears that the nighttime
525 chemistry is poorly represented in the first layer of GEOS-Chem, the causes of which warrant further investigations.

## 5. Conclusions and discussion

This study uses a newly compiled EOF-EEMD analysis-visualization package to evaluate the spatiotemporal variations of hourly $NO_2$ and $PM_{2.5}$ data over Eastern China during Fall-
530 Winter 2013. The observed $NO_2$ data exhibit an Eastern China-wide synchronous pattern (EOF1) and a north-south contrasting pattern (EOF2). EOF1 of $NO_2$ consists of a dominant signal for diurnal variation and a weaker signal for day-to-day variation. EOF2 of $NO_2$ is dominated by the day-to-day variation. Although the diurnal cycle is relatively consistent across the days, the day-to-day variation exhibits a RPR at 2–5 days with no constant
535 amplitude, a feature intended to be properly accounted for in the EOF-EEMD analysis. The day-to-day variation is largely driven by cold air passage, as revealed from analyses of observed wind speed, temperature, and RH. In particular, wind speed is most closely related to $NO_2$, based on an EOF-EEMD analysis and a complementary correlation calculation (R = -0.77 – -0.73 over NEC and SEC).

540 An EOF-EEMD analysis of the observed $PM_{2.5}$ also reveals an Eastern China-wide synchronous (EOF1) and a north-south contrasting (EOF2) pattern. However, the diurnal variation of $PM_{2.5}$ is much noisier than that of $NO_2$. The day-to-day variation dominates for $PM_{2.5}$, and it is much associated wind speed, especially over NEC (R = -0.60).

Further evaluation on GEOS-Chem and WRF/CMAQ simulations shows that both models
545 simulate the observed EOF1 and EOF2 patterns fairly well. Both models capture the day-to-day variability of $PM_{2.5}$ better than that of $NO_2$. CMAQ outperforms GEOS-Chem in Eastern



China-wide synchronous day-to-day IMFs, especially for $NO_2$; whereas GEOS-Chem better captures the north-south contrasting day-to-day IMFs. CMAQ overestimates the diurnal variability of $NO_2$ and $PM_{2.5}$, such that the IMFs out of the EOF-EEMD analysis are overly dominated by the diurnal signal (especially for $NO_2$). This is likely due to its underestimate of PBL mixing, which deficiencies have been alleviated by the latest model updates. GEOS-Chem underestimates the concentrations of both pollutants, due in part to missing secondary organic aerosols and anthropogenic dusts (affecting $PM_{2.5}$) and a first layer too thick (130 m) to capture the vertical gradient near the ground. GEOS-Chem captures the diurnal variations of $NO_2$ and $PM_{2.5}$. It underestimates the day-to-day variability of nighttime $NO_2$ likely due to chemical inaccuracies in the first layer.

This study suggests that the EOF-EEMD package is a useful tool providing a simultaneous and quantitative view of the spatial and temporal (both stationary and non-stationary) scales embedded in a dataset. The package can be applied to other chemical, meteorological or climatic variables, and will be freely accessible to the public.

### Acknowledgments

This research is supported by the National Natural Science Foundation of China (41775115) and the 973 program (2014CB441303). Air pollution observations are taken from the Ministry of Environmental Protection (http://106.37.208.233:20035/). Meteorological measurements are taken from the NOAA NCEI (http://gis.ncdc.noaa.gov/map/viewer/#app=clim&cfg=cdo&theme=hourly&layers=1&node=gis).

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





Table 1. Correlation between PCs and regional mean values in terms of diurnal and day-to-day variability for $NO_2$.

| PC / Region | PC1 | | PC2 |
|---|---|---|---|
| | Diurnal | Day-to-day | Day-to-day |
| Eastern China (obs.) | 1.0** | 0.96** | 0.0 |
| NEC (obs.) | 1.0** | 0.77** | 0.66** |
| SEC (obs.) | 1.0** | 0.84** | -0.45** |
| Eastern China (GEOS-Chem) | 1.0** | 0.97** | -0.07 |
| NEC (GEOS-Chem) | 1.0** | 0.56** | 0.81** |
| SEC (GEOS-Chem) | 1.0** | 0.78** | -0.47** |
| Eastern China (CMAQ) | 1.0** | 0.99 ** | 0.12 |
| NEC (CMAQ) | 1.0** | 0.82** | 0.74** |
| SEC (CMAQ) | 1.0** | 0.94** | -0.32** |

** The correlation coefficient is statistically significant with the P-value < 0.01.




Table 2. Correlation between PCs and regional mean values in terms of diurnal and day-to-day variability for PM$_{2.5}$.

| PC / Region | PC1 | | PC2 |
|---|---|---|---|
| | Diurnal | Day-to-day | Day-to-day |
| Eastern China (obs.) | 0.99** | 0.97** | -0.23 |
| NEC (obs.) | 0.99** | 0.89** | 0.41** |
| SEC (obs.) | 0.99** | 0.78** | -0.62** |
| Eastern China (GEOS-Chem) | 1.0** | 0.98** | -0.13 |
| NEC (GEOS-Chem) | 1.0** | 0.85** | 0.55** |
| SEC (GEOS-Chem) | 1.0** | 0.72** | -0.68** |
| Eastern China (CMAQ) | 1.0** | 0.99** | -0.20 |
| NEC (CMAQ) | 1.0** | 0.89** | 0.32** |
| SEC (CMAQ) | 1.0** | 0.90** | -0.62** |

** The correlation coefficient is statistically significant with the P-value < 0.01.




Table 3. Observed and simulated pollutants and their correlations.

| | | NEC | | | SEC | | |
|---|---|---|---|---|---|---|---|
| | | Mean | Median | R [1] | Mean | Median | R [1] |
| NO$_2$ (hourly) | Observation | 62.4 | 62.3 | / | 56.0 | 55.8 | / |
| | GEOS-Chem | 41.0 | 41.5 | 0.96** | 43.3 | 45.7 | 0.96** |
| | R_GEOS-Chem [2] | 50.7 | 52.3 | 0.96** | 54.9 | 57.5 | 0.96** |
| | CMAQ | 78.4 | 79.4 | 0.94** | 68.7 | 68.3 | 0.95** |
| NO$_2$ (daily mean) | Observation | 62.4 | 65.2 | / | 56.0 | 57.2 | / |
| | GEOS-Chem | 41.0 | 40.4 | 0.25* | 43.3 | 43.0 | 0.37** |
| | R_GEOS-Chem [2] | 50.7 | 51.9 | 0.24 | 54.9 | 52.6 | 0.29* |
| | CMAQ | 78.4 | 79.4 | 0.84** | 68.8 | 67.0 | 0.63** |
| PM$_{2.5}$ (hourly) | Observation | 92.1 | 95.1 | / | 111.4 | 115.6 | / |
| | GEOS-Chem | 61.1 | 65.6 | 0.83** | 69.8 | 74.7 | 0.86** |
| | R_GEOS-Chem [2] | 88.4 | 104.0 | 0.76** | 81.9 | 92.2 | 0.80** |
| | CMAQ | 130.4 | 144.3 | 0.81** | 135.4 | 144.1 | 0.81** |
| PM$_{2.5}$ (daily mean) | Observation | 92.1 | 90.6 | / | 111.4 | 111.7 | / |
| | GEOS-Chem | 61.1 | 56.3 | 0.75** | 69.8 | 63.5 | 0.55** |
| | R_GEOS-Chem [2] | 88.4 | 81.8 | 0.75** | 81.9 | 76.4 | 0.56** |
| | CMAQ | 130.4 | 128.6 | 0.87** | 135.4 | 128.0 | 0.88** |

1. Correlation between observed and simulated variables. ** indicates the correlation coefficient is statistically significant with the P-value < 0.01 while * indicates it passed statistical test with P-value < 0.01.

2. Revised GEOS-Chem NO$_2$ and PM$_{2.5}$ by multiplying the ratio of the first layer to the second layer of CMAQ values.







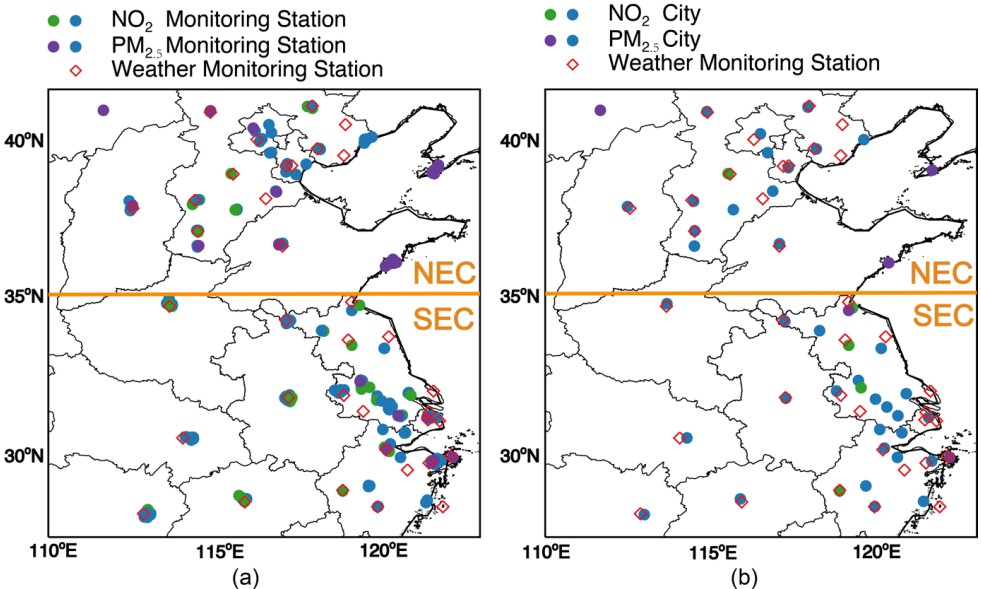


Figure 1. (a) Distribution of 163 measurement stations for NO$_2$, 157 stations for PM$_{2.5}$ and 36 meteorological stations (red diamonds) over Eastern China (25°N–41°N, 110°E–123°E). (b) Distribution of 42 cities with NO$_2$ and PM$_{2.5}$ observations. Both dots denote stations (a) and cities (b) with both NO$_2$ and PM$_{2.5}$ data. The blue dots indicates the same stations (cities) for both NO$_2$ and PM$_{2.5}$ while the green dots only used for NO$_2$ and purple dots only used for PM$_{2.5}$. The orange line separates Northern Eastern China (NEC) and Southern Eastern China (SEC).




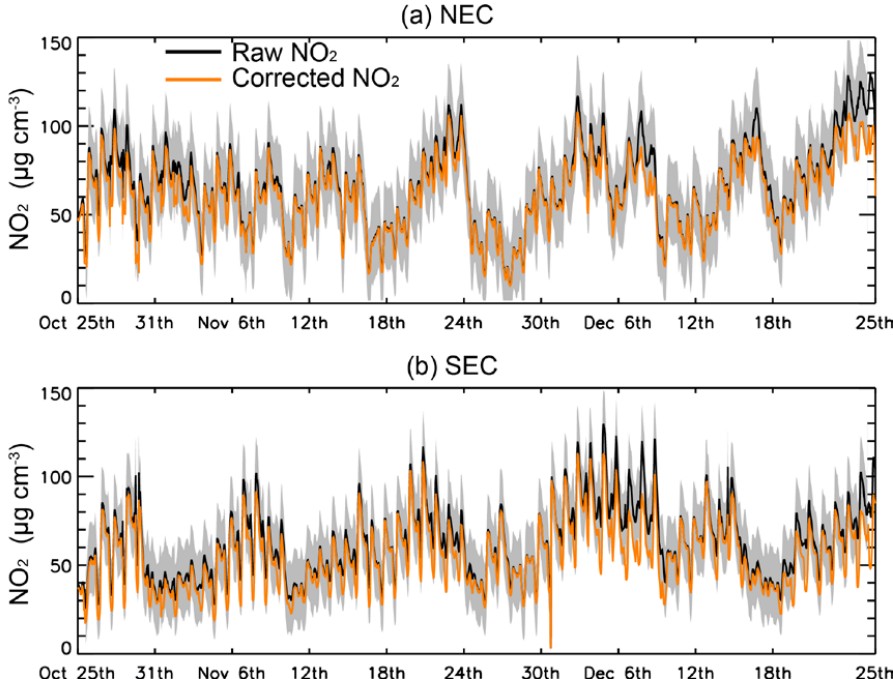


Figure 2. Regional mean hourly time series of raw and "corrected" $NO_2$ from the observations. The gray shading indicates one standard deviation across all stations.




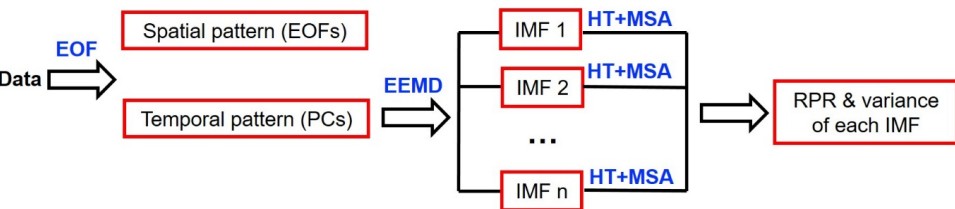

Figure 3. The flow chart of EOF-EEMD analysis-visualization package. The red boxes represent quantities visualized.






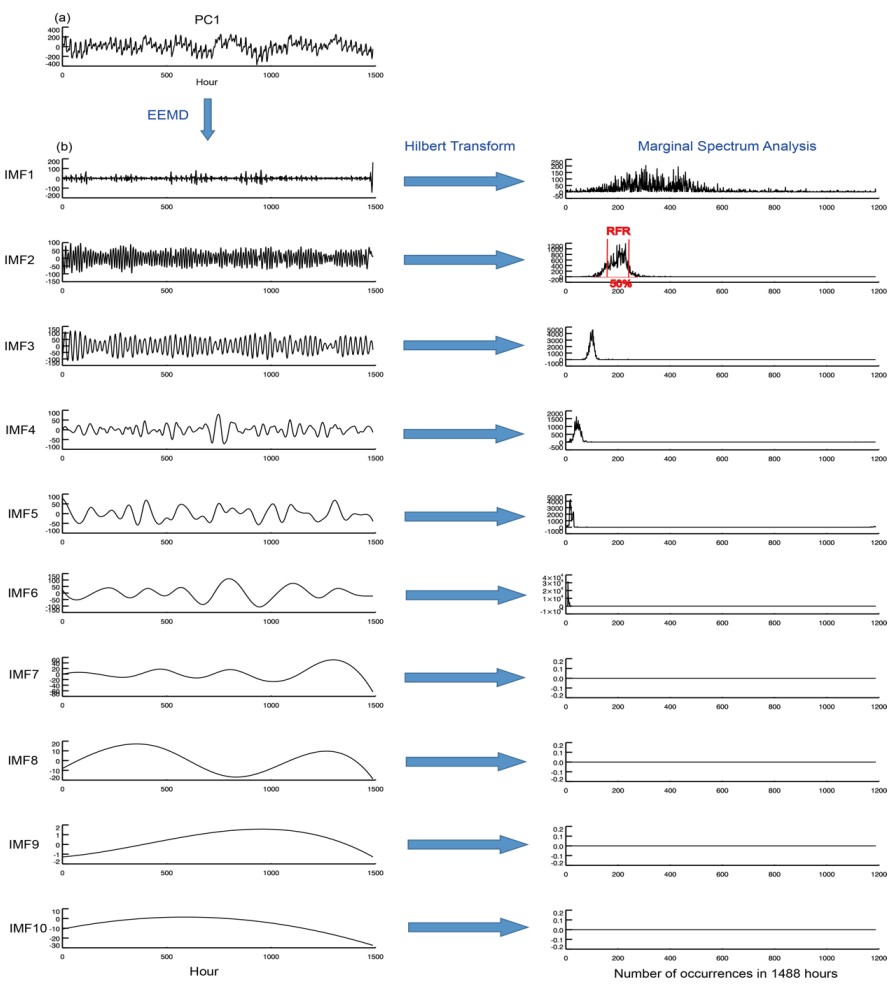

Figure 4. EEMD-HT-MSA result for PC1 of observed NO₂.






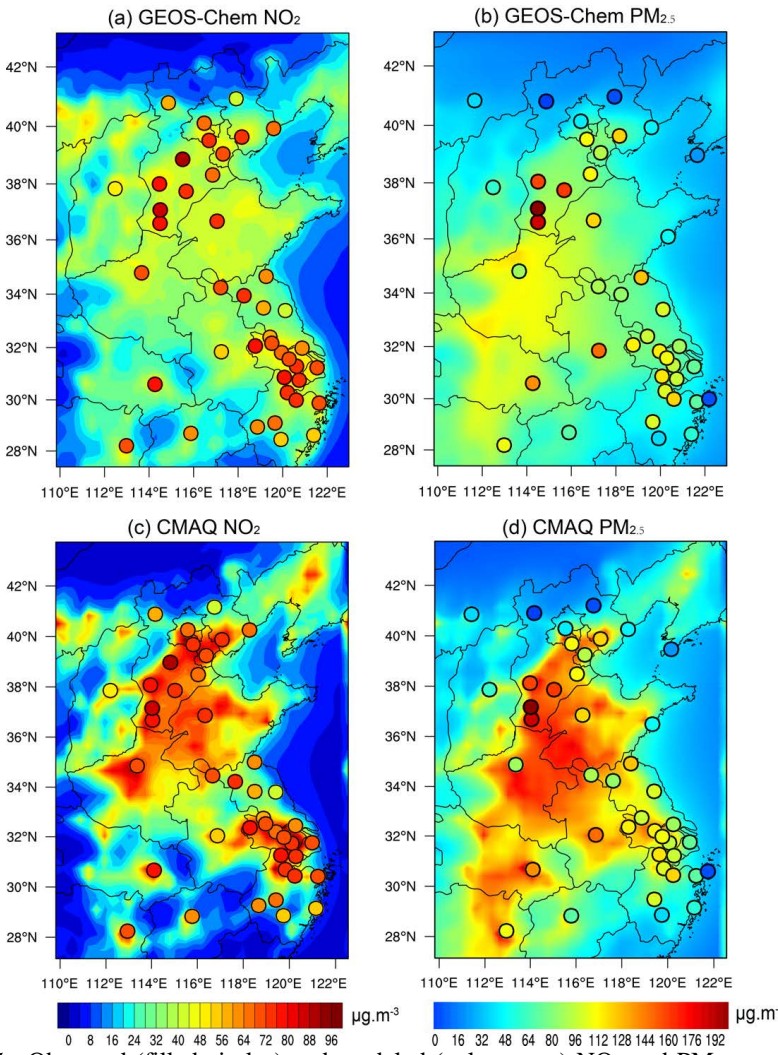

Figure 5. Observed (filled circles) and modeled (color maps) $NO_2$ and $PM_{2.5}$ averaged over October $25^{th}$ – December $25^{th}$ 2013. Here the model results are averaged over all days rather than sampled at times of valid observations.






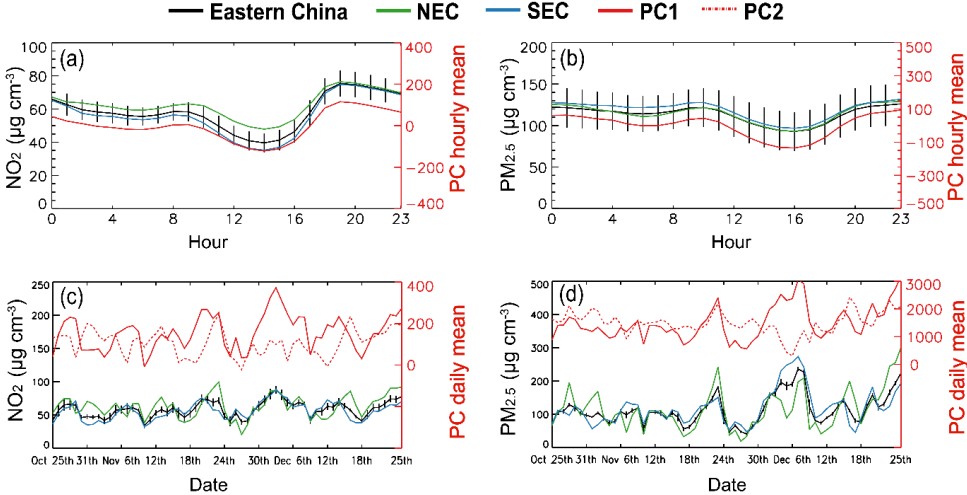

Figure 6. (a) Diurnal variation of observed $NO_2$ averaged over October $25^{th}$ – December $25^{th}$ 2013. The black vertical bars represent one standard deviation across the days. PC1 from the EOF analysis is overlaid in red. (b) Similar to (a) but for $PM_{2.5}$. (c) Day-to-day variation of daily mean $NO_2$ over October $25^{th}$ – December $25^{th}$ 2013. Data are de-trended. The black vertical bars represent one standard deviation due to the diurnal variation. PC1 and PC2 from the EOF analysis are overlaid in red. (d) Similar to (c) but for $PM_{2.5}$.



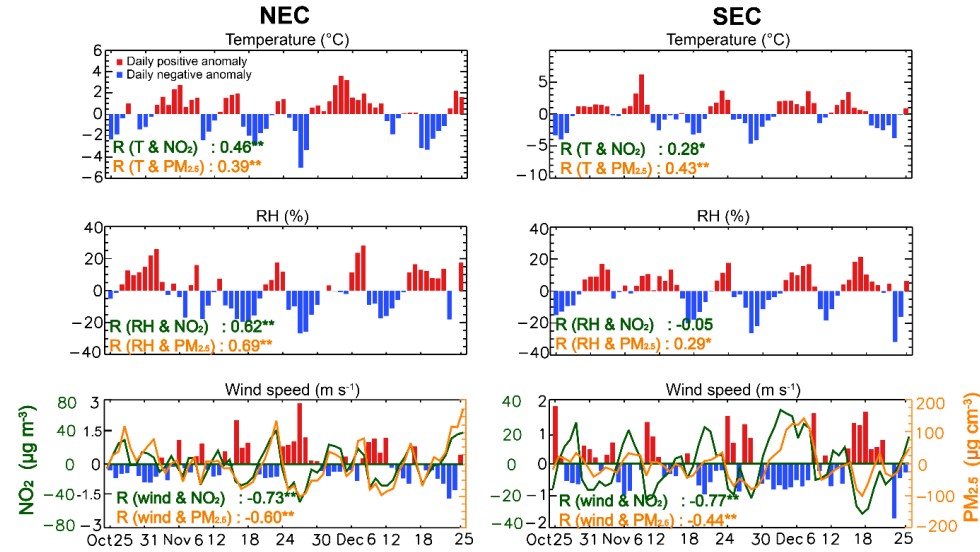


Figure 7. Daily anomalies of observed meteorological parameters and pollutant concentrations averaged over NEC and SEC, as well as their correlations. All data are detrended. Correlation coefficients with "*" and "**" are statistically significant with P-value below 0.05 and 0.01, respectively.




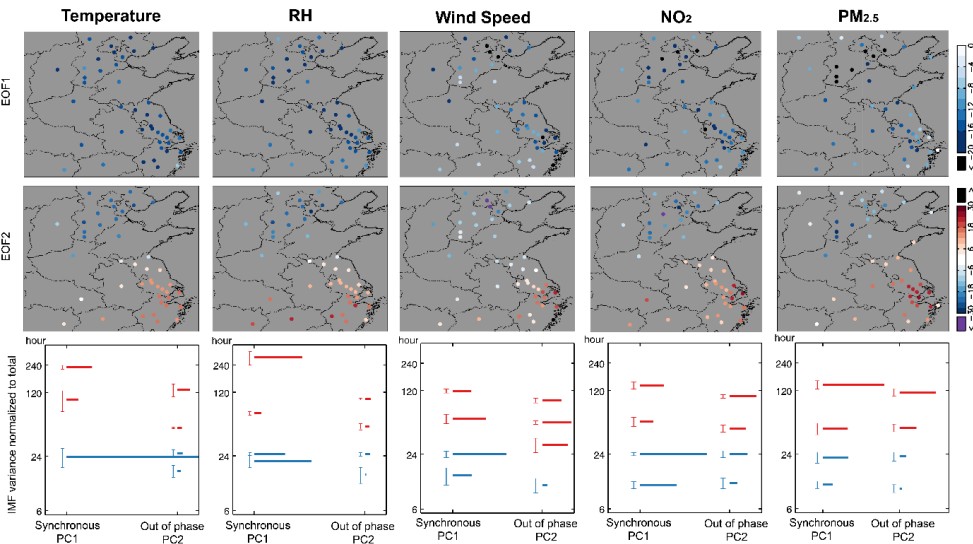

Figure 8. EOF-EEMD-HT-MSA results for the observed temperature, RH, wind speed, $NO_2$ and $PM_{2.5}$. The first two rows depict EOF1 and EOF2, and the third row shows the EEMD-HT-MSA result for PC1 and PC2. In each panel of the third row, the length of the vertical "error bar" shows the RPR of an IMF, while the length of the horizontal bar represents the percentage contribution of the IMF to the total variance of the original data (as such, the horizontal lengths for different IMFs across different PCs can be compared). The blue (red) color indicates diurnal (day-to-day) variation.




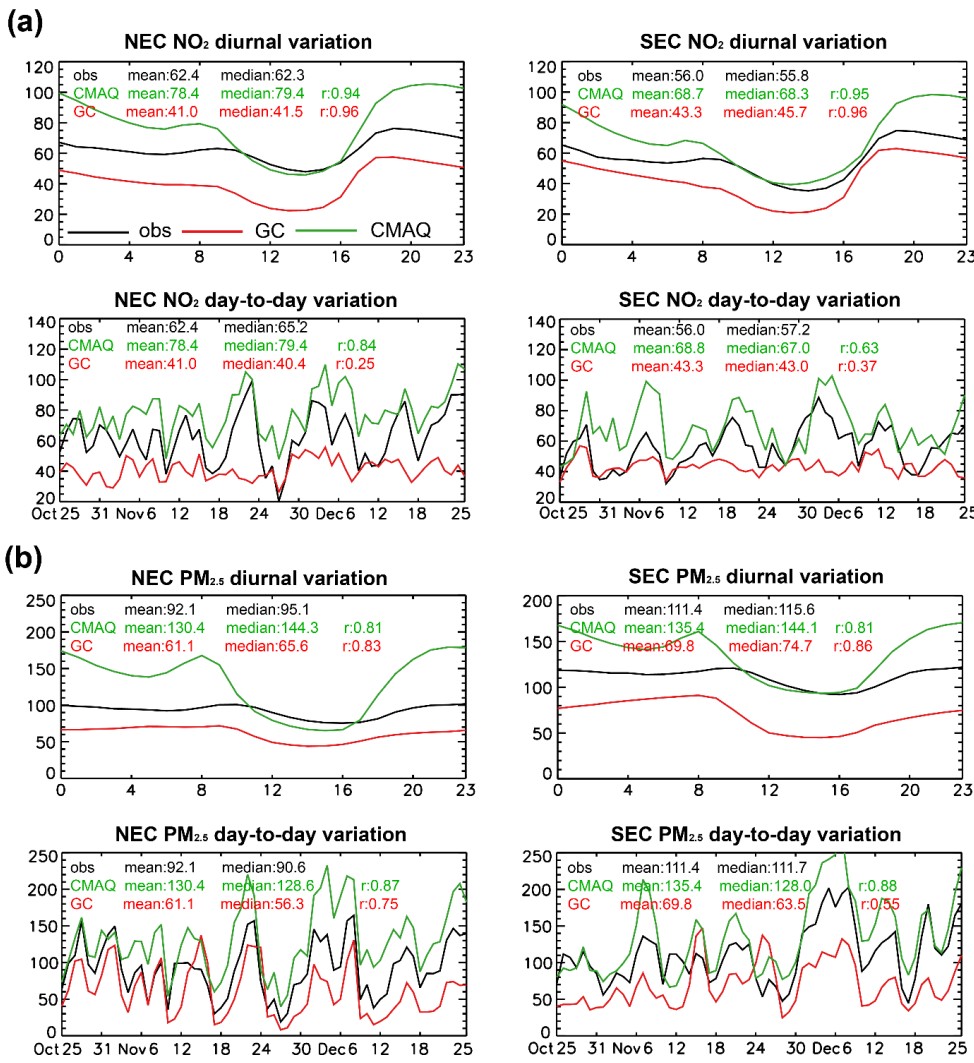

Figure 9. Observed and simulated diurnal and day-to-day variations of (a) NO$_2$ and (b) PM$_{2.5}$ over NEC and SEC ($\mu$ g m$^{-3}$).





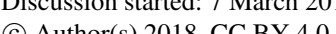

Figure 10. EOF-EEMD-HT-MSA results for observed, GEOS-Chem and CMAQ NO₂. See Sect. 4.2 for detailed descriptions.





Figure 11. EOF-EEMD results for observed, GEOS-Chem and CMAQ PM₂.₅. See Sect. 4.2 for detailed descriptions.





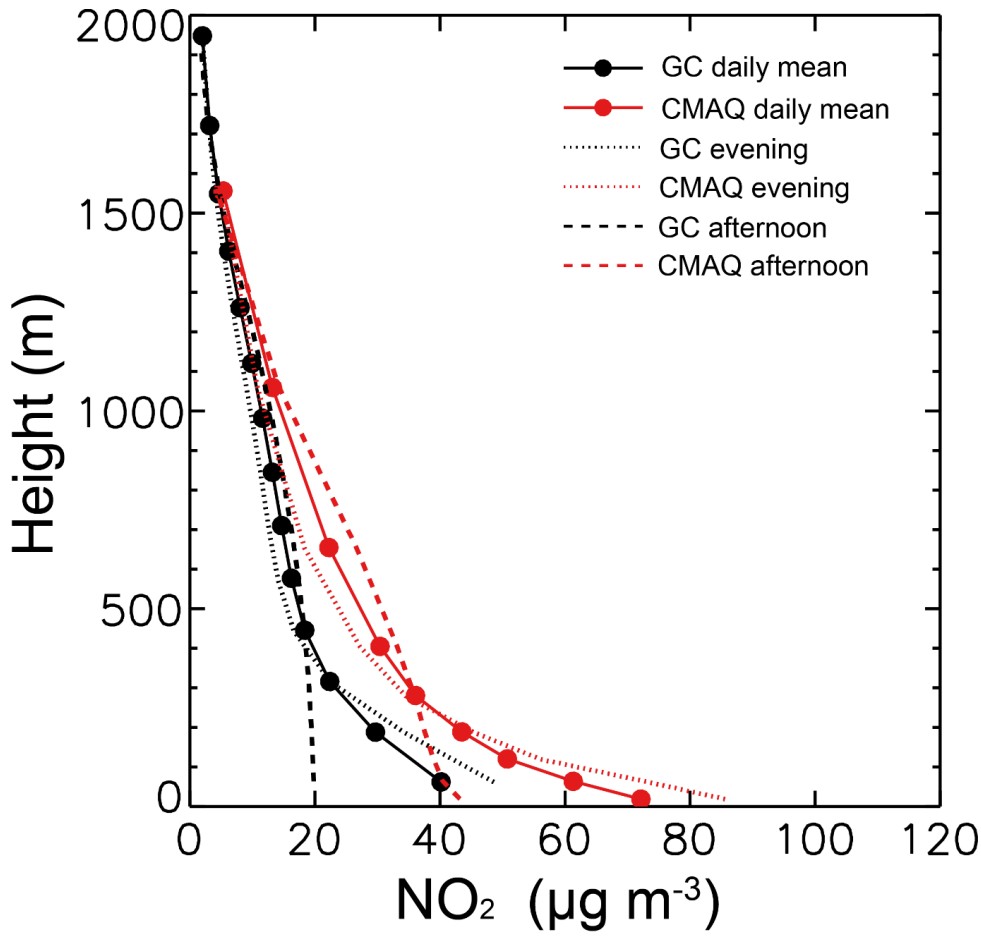

Figure 12. Eastern China mean NO$_2$ vertical profiles simulated by GEOS-Chem and CMAQ
averaged over October 25$^{th}$ – December 25$^{th}$ 2013. The black and red dots denote the center
of each vertical layer in the two models. The evening is from 20:00 to 23:00 LT while the
afternoon is from 12:00 to 15:00 LT.

865