# Peer review of "Spatiotemporal variability of NO2 and PM2.5 over Eastern China: observational and model analyses with a novel statistical method"

_Atmospheric Chemistry and Physics, 2017_

## Referee Comment (RC1) · Anonymous Referee #2 · 1 Apr 2018

In this paper, the authors analyzed the spatial and temporal variability of ground level NO2 and PM2.5 in Oct-Dec. 2013, and evaluated model performance of GEOS-Chem and CMAQ on the spatial and temporal variability. The topic is important, the method is sound, and the results look reasonable. I suggest this manuscript be accepted as a discussion paper with some minor revision described below.

Revision suggestion: (1) The separation of SEC and NEC using Huai-River would be more appropriate, especially when considering the variability of NO2 and PM2.5 due to emissions and meteorology factors.

(2) The emissions used in GEOS-Chem and CMAQ are different and it adds more

complexity to illustrate the performance difference between the two models. Better to use the same emissions to eliminate this factor, or at least to discuss how this factor contributed to the difference

(3) If the too thick first layer of GEOS-Chem (130m) is the main reason for model underprediction, is it possible to configure the first layer to 80m as the CMAQ model so that you can provide direct proof to support your argument?

---

## Referee Comment (RC2) · Anonymous Referee #1 · 22 May 2018

This paper analyzes spatial and temporal variability in Fall-Winter 2013 of PM2.5 and NO2 in Eastern China in observations, GEOS-Chem, and CMAQ. The authors have done a good job of distilling a lot of complex information in a clear format. The EOF-EEMD method seems novel and is interesting. I recommend acceptance after the following issues are addressed.

The most substantial issue is the choice to exclude SOA from the GEOS-Chem model simulations. This is a huge caveat that strongly impacts the interpretation of any of the GEOS-Chem results and conclusions. The authors correctly point out that more than 20% of the PM2.5 in Eastern China could be from SOA. The reason given for not

including SOA is that the mechanism underestimates aerosol formation, but then the main conclusion about GEOS-Chem is that it severely underpredicts PM2.5 in Eastern China anyway, so this is circular reasoning. Even a simple SOA scheme would be better than nothing. I recommend that the authors add some discussion about how the lack of SOA may impact their evaluation of GEOS-Chem, beyond just the bias in the seasonal mean. How does the lack of SOA impact the day-to-day and diurnal model-observation comparison? Since it is stated that CMAQ includes SOA, perhaps you can draw conclusions about the impact of SOA from CMAQ.

(By the way, it is stated in the manuscript that GEOS-Chem "does not include" SOA, but the more accurate statement would be something like "these simulations of GEOS-Chem do not include SOA", since the capability is there in the model in general.)

Minor comments: Line 33-34. Mention what time averaging frame these biases refer to (I assume Fall-Winter seasonal average)

Line 39 and several other places in the manuscript: change "anthropogenic dusts" to "anthropogenic dust". Also there was a mention of "sea salts" which should be changed to "sea salt". These are typically already plural without the added "s".

Line 50. "Downwind" is probably more appropriate than "downstream" since we're talking about air pollution here.

Line 81. PM2.5 is called "the dominant pollutant". In terms of what? Emissions, mass, human health burden? Specify and cite.

Line 86. Reference Wu et al. is written in all capital letters. Same in the reference list. Please fix.

Line 93. I don't know what the "M" in "MEEMD" is. Thus far, I think you've only defined the acronym "EEMD"

Line 277. "while noise" should be "white noise"

[Figure]

Section 3.1 and Figure 5. There is a units issue here. The text says microgram per cubic meter, but Figure 6 is labeled as microgram per cubic centimeter. Surely that must be an error. Please check all units.

Section 3.2. Why are only RH, wind, and temperature chosen for meteorological parameters? Was this the only data available. Justify the choice.

Line 545: "...both models simulate the observed EOF1 and EOF2 patterns fairly well". This is vague. "Fairly well" is a value judgement; please quantify.

Figure 9 and throughout: Label your plot axes.

---

## Author Comment (AC1) · 21 Jun 2018

In this paper, the authors analyzed the spatial and temporal variability of ground level NO2 and PM2.5 in Oct-Dec. 2013, and evaluated model performance of GEOS-Chem and CMAQ on the spatial and temporal variability. The topic is important, the methods sound, and the results look reasonable. I suggest this manuscript be accepted as a discussion paper with some minor revision described below.

(1) The separation of SEC and NEC using Huai-River would be more appropriate, especially when considering the variability of NO2 and PM2.5 due to emissions and meteorology factors.

[Figure]

Our separation line is based on the EOF analysis. The line is also close to the Huaihe River line (red line in the updated Fig. 1), especially considering that few measurement stations are located between the two lines. Using the Huaihe River line does not affect our general findings regarding south-north contrast.

(2) The emissions used in GEOS-Chem and CMAQ are different and it adds more complexity to illustrate the performance difference between the two models. Better to use the same emissions to eliminate this factor, or at least to discuss how this factor contributes to the difference.

In the end of revised Sect. 4.3, we have added that "The magnitude of emission differences between the two models plays an insignificant role in the differences between their simulated $NO_2$ or $PM_{2.5}$ concentrations. Chinese anthropogenic emissions in 2010 used in GEOS-Chem (except for $NO_x$) are close to emissions in 2013 used in CMAQ (within 10% for both gases and primary aerosols, mostly within 5%, see Zheng et al. (2018)). $NO_x$ emissions in GEOS-Chem are scaled to 2013 using satellite $NO_2$ data, which further eliminates the differences from those used in CMAQ. The difference in the spatial distribution of emissions is also small (Geng et al., 2017; Zheng et al., 2018)."

(3) If the too thick first layer of GEOS-Chem (130m) is the main reason for model underprediction, is it possible to configure the first layer to 80m as the CMAQ model so that you can provide direct proof to support your argument?

This is a very good suggestion. In fact, we had thought about doing so. Unfortunately, the vertical resolution of GEOS-Chem is hardwired and adhered to the coordinate of inputted meteorological fields, unlike other models such as MOZART. Thus it would take much longer time to change the model coordinate than would be appropriate for this particular study. This was why we have taken the liberty to use CMAQ simulated vertical profiles to scale GEOS-Chem results, as a simplified test.

---

## Author Comment (AC2) · 21 Jun 2018

The most substantial issue is the choice to exclude SOA from the GEOS-Chem model simulations. This is a huge caveat that strongly impacts the interpretation of any of the GEOS-Chem results and conclusions. The authors correctly point out that more than 20% of the PM2.5 in Eastern China could be from SOA. The reason given for not including SOA is that the mechanism underestimates aerosol formation, but then the main conclusion about GEOS-Chem is that it severely under predicts PM2.5 in Eastern China anyway, so this is circular reasoning. Even a simple SOA scheme would be better than nothing. I recommend that the authors add some discussion about

how the lack of SOA may impact their evaluation of GEOS-Chem, beyond just the bias in the seasonal mean. How does the lack of SOA impact the day-to-day and diurnal model observation comparison? Since it is stated that CMAQ includes SOA, perhaps you can draw conclusions about the impact of SOA from CMAQ. (By the way, it is stated in the manuscript that GEOS-Chem "does not include" SOA, but the more accurate statement would be something like "these simulations of GEOS-Chem do not include SOA", since the capability is there in the model in general.) This simulation of GEOS-Chem underestimates PM2.5 by about 35% (Table 3), much larger than can be explained by the missing SOA (up to 21%). Thus we suggested in the original manuscript that other factors may also play some roles.

We have also revised the text to indicate that this simulation of GEOS-Chem does not include SOA. As added in the end of revised Sect. 4.3: "We further use CMAQ simulations to investigate whether the inclusion of SOA affects our analysis of the spatiotemporal patterns of PM2.5. Supplementary Fig. S1 compares the time series of CMAQ-simulated PM2.5 with versus without including SOA. Although SOA contributes about 8-9 $\mu$g/mˆ3 of PM2.5 averaged over the days, inclusion of SOA does not affect the temporal variability. The EOF-EEMD results in Supplementary Fig. S2 further confirm that the spatiotemporal scales are very consistent whether or not SOA is included."

Minor comments:

Line 33-34. Mention what time averaging frame these biases refer to (I assume Fall-Winter seasonal average)

Changed.

Line 39 and several other places in the manuscript: change "anthropogenic dusts" to "anthropogenic dust". Also there was a mention of "sea salts" which should be changed to "sea salt". These are typically already plural without the added "s".

Changed.

Line 50. "Downwind" is probably more appropriate than "downstream" since we're talking about air pollution here.

Changed.

Line 81. PM2.5 is called "the dominant pollutant". In terms of what? Emissions, mass, human health burden? Specify and cite.

Changed and relevant citations added.

Line 86. Reference Wu et al. is written in all capital letters. Same in the reference list. Please fix.

Changed.

Line 93. I don't know what the "M" in "MEEMD" is. Thus far, I think you've only defined the acronym "EEMD"

Changed.

Line 277. "while noise" should be "white noise"

Changed.

Section 3.1 and Figure 5. There is a units issue here. The text says microgram per cubic meter, but Figure 6 is labeled as microgram per cubic centimeter. Surely that must be an error. Please check all units.

The units labeled on Figure 6 were wrong, and we have corrected them.

Section 3.2. Why are only RH, wind, and temperature chosen for meteorological parameters? Was this the only data available. Justify the choice.

We chose these meteorological parameters based on previous studies (line 65-67). Three-hourly data for surface pressure, temperature, RH and wind speed are available from the meteorological stations. We do not use surface pressure additionally, because it is highly correlated to air temperature and relative humidity on the day-to-day scale.

Line 545: "...both models simulate the observed EOF1 and EOF2 patterns fairly well". This is vague. "Fairly well" is a value judgement; please quantify.

Changed.

Figure 9 and throughout: Label your plot axes.

We have added units in the caption.